# Family of Deep Image Prior Networks for Accelerated 3D LGE-MRI Acquisition with Enhanced Reconstruction

**Md Hasibul Husain Hisham**[1]                                              HASIBUL.HISHAM@UTAH.EDU
**Shireen Elhabian**[1]                                                            SHIREEN@SCI.UTAH.EDU
**Ganesh Adluru**[2]                                          GANESHSHARMA.ADLURU@HSC.UTAH.EDU
**Andrew Arai**[2,3]                                                    ANDREW.ARAI@HSC.UTAH.EDU
**Eugene Kholmovski**[2]                                  EVGUENI.KHOLMOVSKI@HSC.UTAH.EDU
**Ravi Ranjan**[3]                                                    RAVI.RANJAN@HSC.UTAH.EDU
**Edward DiBella**[2]                                          EDWARD.DIBELLA@HSC.UTAH.EDU

[1] *Kahlert School of Computing, University of Utah, Salt Lake City, UT, United States*

[2] *Radiology & Imaging Sciences, University of Utah, Salt Lake City, UT, United States*

[3] *Cardiology, University of Utah, Salt Lake City, UT, United States*

**Editors:** Accepted for publication at MIDL 2025

## Abstract

Late Gadolinium Enhancement (LGE) MRI is essential for visualizing and treating left atrial fibrosis, but current protocols require lengthy acquisition times (7-20 minutes) and often produce suboptimal image quality. While recent advances in isotropic imaging have shown promise, scan times of 12-15 minutes still present clinical challenges. This study evaluates the efficacy of existing Deep Image Prior (DIP) frameworks for accelerated 3D LGE-MRI reconstruction. We comprehensively assess multiple DIP variants - vanilla DIP, reference-guided DIP, DIP with Total Variation, and self-guided DIP - on their ability to reconstruct high-quality isotropic ($1.25mm^3$) images from highly undersampled k-space data. Using data from 10 subjects, we demonstrate that self-guided DIP achieves superior reconstruction quality (PSNR: $32.8\pm1.2$ dB, SSIM: $0.891\pm0.015$ at 1/4th of acquisition time) compared to traditional compressed sensing and other DIP variants. Our evaluation shows that DIP-based reconstruction can maintain diagnostic quality with acquisition times reduced to 2-4 minutes, particularly in preserving thin left atrial wall details. These findings suggest that DIP-based methods could improve clinical workflow efficiency and patient comfort in high-resolution 3D LGE studies for atrial fibrillation patients.

**Keywords:** MRI Image Acquision & Reconstruction, Deep Image Prior, Unsupervised Learning

## 1. Introduction

LGE-MRI plays a role in the clinical management of atrial fibrillation patients (McGann et al., 2008). Current 3D LGE-MRI protocols, while effective, face significant challenges that limit their clinical utility. The standard ECG-gated and respiratory navigated sequences require lengthy acquisition times (typically 7-20 minutes), often resulting in patient discomfort and increased susceptibility to motion artifacts. Moreover, these sequences frequently produce inadequate image quality and suffer from anisotropic resolution limitations. The clinical need for high-resolution isotropic imaging is particularly acute in cardiac applications, where precise visualization of thin atrial walls is crucial for diagnosis and treatment planning.

While recent advances in fixed-time isotropic 3D LGE-MRI methods have shown promise in improving spatial resolution, the current standard of 12-15 minute acquisition times still presents challenges for patient comfort and workflow efficiency. Additionally, the prevalence of respiratory motion in these scans introduces unique complexities not encountered in other anatomical imaging sites, such as brain or knee imaging. This research studies the efficacy to accelerate 3D LGE-MRI reconstruction using a family of Deep Image Prior (DIP) methods. Unlike traditional supervised deep learning approaches that require extensive training data, DIP-based methods offer several distinct advantages for cardiac imaging:

- They optimize network parameters for each specific case, allowing adaptation to patient-specific characteristics such as unique anatomical variations, motion patterns, and image contrast properties. This personalization is particularly valuable in cardiac imaging where inter-patient variability is high.

- They maintain robust performance even with highly undersampled K-space data by leveraging the network architecture itself as a learned prior. This approach naturally handles the incoherent artifacts that arise from undersampling, making it particularly effective for accelerated acquisitions.

- They avoid the need for large training datasets, which are particularly challenging to obtain in cardiac imaging due to the complexity of acquiring fully-sampled reference data and motion-related challenges.

This work aims to achieve high-quality isotropic reconstructions ($1.25\text{mm}^3$) from significantly reduced acquisition times (approximately 2-4 minutes), representing a 4-6X acceleration compared to current standards. The goal is to maintain diagnostic quality while addressing the fundamental limitations of current protocols, which could expand the accessibility of 3D LGE studies for atrial fibrillation patients.

## 2. Related Works

The field of accelerated MRI reconstruction has evolved significantly over the past decade, progressing from traditional compressed sensing approaches to sophisticated deep learning methods. Compressed sensing (CS) leverages the inherent sparsity of MRI data in appropriate transform domains to enable high-quality reconstruction from undersampled K-space measurements. Such approaches usually combine data consistency terms with carefully chosen regularizers, such as Total Variation (TV) (Ehrhardt and Betcke, 2016; Chen et al., 2014) or wavelets (Guerquin-Kern et al., 2011; Lai et al., 2016), to promote sparsity while preserving image features. Ganesh et al.'s recent works include combining the above steps with Block-Matching and 3D filtering (BM3D), thus exploiting non-local self-similarity in images (Adluru et al., 2022). However, these iterative reconstruction methods often require careful parameter tuning and can be computationally intensive, especially for high-resolution 3D acquisitions.

Early Deep Learning-based approaches focused on supervised learning, where networks were trained on paired datasets of undersampled and fully-sampled images (Ramanarayanan et al., 2023). While effective, these methods face limitations in clinical deployment due to

their dependence on high-quality training data and potential challenges in generalizing to different acquisition protocols or anatomical variations. Ulyanov et al. proposed Deep Image Prior (DIP) which bridges the gap between traditional optimization-based methods and deep learning (Ulyanov et al., 2018). DIP leverages the structure of convolutional neural networks as an implicit prior, eliminating the need for training data. The framework has been successfully adapted to various image restoration tasks, including super-resolution, inpainting, and denoising (Jo et al., 2021).

In the context of MRI reconstruction, several DIP variants have been proposed. Yazdan-panah et al. have adapted the basic DIP framework to the MRI physics model (Pour Yazdanpanah et al., 2019). Xue et al. have explored combining DIP with TV to leverage both the implicit network prior and explicit spatial regularization, on 3D CMRA data (Xue et al., 2024). However, they perform the DIP in a slice-by-slice manner (2D), not utilizing the 3D resolution. Building on these developments, self-guided learning has emerged as a promising direction in unsupervised image reconstruction (Liang et al., 2024). Bell et al. have leveraged the network's predictions as a form of regularization, reducing the dependence on hand-crafted priors (Bell et al., 2023).

Late Gadolinium Enhancement (LGE) cardiac MRI presents unique challenges that make it an ideal testing ground for advanced reconstruction methods. The need to capture fine structural details in the heart while managing respiratory motion makes high-resolution isotropic imaging particularly challenging. Traditional approaches often rely on anisotropic acquisitions, sacrificing through-plane resolution for reduced scan time. Recent work has focused on enabling high-quality isotropic imaging through advanced reconstruction techniques (Adluru et al., 2022), though maintaining diagnostic quality with highly accelerated acquisitions remains an open challenge.

## 3. Data Acquisition

We evaluated our methods on data from 10 human subjects acquired using Siemens 3T MRI scanners with 20-channel cardiac coils. The imaging protocol used a segmented inversion-recovery gradient echo sequence optimized for cardiac imaging, achieving an isotropic 1.25 mm$^3$ resolution with 64-140 axial slices. Sequence parameters included TR = 2.7 ms, TE = 1.5 ms, with individually optimized Inversion Time (TI) determined via a TI scout sequence for optimal myocardial nulling. Motion compensation was implemented through ECG-gating that targets the diastolic phase and a one-dimensional navigator at the liver-diaphragm interface. The number of phase-encoding lines was picked at the start of the acquisition based on the heartrate.

For our isotropic acquisitions, the K-space data was collected continuously regardless of the navigator position. The 3D K-space was filled using a spiral-like variable density sampling pattern with golden ratio-based angular and radial spacing, with strong oversampling in the central ky-kz region. All acquisitions followed institutional review board-approved protocols with informed consent. This dataset allows evaluation of reconstruction quality at different acceleration rates while maintaining the challenging conditions of clinical cardiac imaging.

## 4. Methodology

### 4.1. Compressed Sensing Reconstruction

The gold standard reconstruction used in this study employs an iterative compressed sensing (CS) approach (Adluru et al., 2022), which solves the following optimization problem:

$$\arg \min_x \left\{ \lambda_1 \|Ax - d\|_2^2 + \lambda_2 \text{TV}(x) + \lambda_3 \text{BM3D}(x) \right\} \tag{1}$$

where $x$ is the reconstructed image, $A$ represents the forward model incorporating coil sensitivities and Fourier transform, $d$ is the acquired K-space data, $\text{TV}(x)$ enforces total variation regularization, and $\text{BM3D}(x)$ applies Block-Matching and 3D filtering. The weights $\lambda_1$, $\lambda_2$, and $\lambda_3$ are carefully tuned using hyperparameter grid search. $\lambda_1$ controls the data consistency with acquired $k$-space measurements, $\lambda_2$ controls the spatial smoothness while preserving edges, and $\lambda_3$ controls the contribution of non-local self-similarity regularization.

### 4.2. Deep Image Prior: Core Concept

The DIP framework introduces a fundamentally different approach to image reconstruction by using an untrained neural network as a parameterized prior. The core idea can be expressed through the following optimization (Ulyanov et al., 2018):

$$\arg \min_\theta \|f_\theta(z) - \tilde{x}\|_2^2 \tag{2}$$

where $f_\theta$ represents a convolutional neural network with parameters $\theta$, $z$ is a fixed random input, and $\tilde{x}$ is the degraded image.

### 4.3. DIP for MRI Reconstruction

Adapting the DIP framework to MRI reconstruction requires incorporating the physics of MRI acquisition. The vanilla DIP formulation for MRI can be expressed as:

$$\arg \min_\theta \sum_{c=1}^{N_c} \|A_c f_\theta(z) - y_c\|_2^2 \tag{3}$$

where $N_c$ represents the number of receiver coils used in the MRI acquisition, $A_c$ represents the forward model including unsdersampling mask, coil sensitivities and the Fourier transform. The network $f_\theta$ learns to map a fixed random noise $z$ to the reconstructed image while maintaining consistency with the acquired K-space data $y_c$ across all coils. This baseline approach, which we refer to as vanilla DIP, serves as the foundation for the subsequent methodological developments.

### 4.4. Reference-Guided DIP

Reference-guided DIP enhances the reconstruction by initializing the input $z$ with zero-filled reconstruction, instead of random noise. Zero-filled reconstruction is the initial image obtained by directly applying inverse Fourier transform to the undersampled k-space data, filling unsampled locations with zeros. The optimization problem maintains the same form as the vanilla DIP. This network conditioning helps guide the optimization process towards more plausible solutions in the early stages of reconstruction (Zhao et al., 2020).

### 4.5. DIP-TV

DIP-TV combines the benefits of both DIP and Total Variation regularization. The optimization problem is formulated as (Liu et al., 2019):

$$\arg\min_{\theta} \sum_{c=1}^{N_c} \|A_c f_\theta(z) - y_c\|_2^2 + \lambda_{\text{TV}} \|\nabla f_\theta(z)\|_1 \tag{4}$$

where $\lambda_{\text{TV}}$ is the TV regularization weight and $\nabla f_\theta(z)$ denotes the spatial gradient of the CNN output, which is computed via finite differences in the image domain. The input z can either be a random noise or zero-filled reconstruction. The total variation term preserves image edges while promoting piecewise smoothness, complementing DIP's ability to capture natural image statistics.

### 4.6. Self-Guided DIP

Self-guided DIP introduces a self-regularization mechanism through network architecture design and optimization strategy. The optimization problem becomes (Bell et al., 2023):

$$\arg\min_{\theta, z} \sum_{c=1}^{N_c} \|A_c \mathbb{E}_\eta[f_\theta(z + \eta)] - y_c\|_2^2 + \alpha \|\mathbb{E}_\eta[f_\theta(z + \eta)] - z\|_2^2 \tag{5}$$

where $\eta$ represents random perturbations, $\mathbb{E}_\eta$ denotes expectation over these perturbations, and $\alpha$ is the weighting parameter that controls the strength of the self-regularization term. The second term acts as a self-guidance mechanism, encouraging the network output to be consistent under small random perturbations of its input. This approach adds stability by acting as an implicit denoising mechanism, effectively preventing overfitting to noise. The final reconstruction is obtained through:

$$x^* = \mathbb{E}_{\eta \sim P_\eta}[f_{\theta^*}(z^* + \eta)] \tag{6}$$

where $\eta$ is sampled from distribution $P_\eta$ (Gaussian in our experiments).

## 5. Results

### 5.1. Implementation Details

The compressed sensing reconstruction was implemented using parallel computing optimization in MATLAB. Through extensive grid-search of the parameters, we set the regularization parameters $\lambda_1 = 0.033$, $\lambda_2 = 8.75 \times 10^{-6}$, and $\lambda_3 = 0.1$, with BM3D denoising parameter $\sigma = 1.25$. After getting the rawdata from the scanner, we retrospectively removed motion-contaminated data by analyzing the navigator signal and retaining only samples acquired during stable respiratory phases. The reconstruction process started with an initial estimate as the zero-filled reconstruction, employing an early stopping criterion with a maximum of 1000 iterations.

For the deep image prior implementations, we utilized a ResUNet architecture with skip connections (Figure 1), which has demonstrated strong performance in medical image

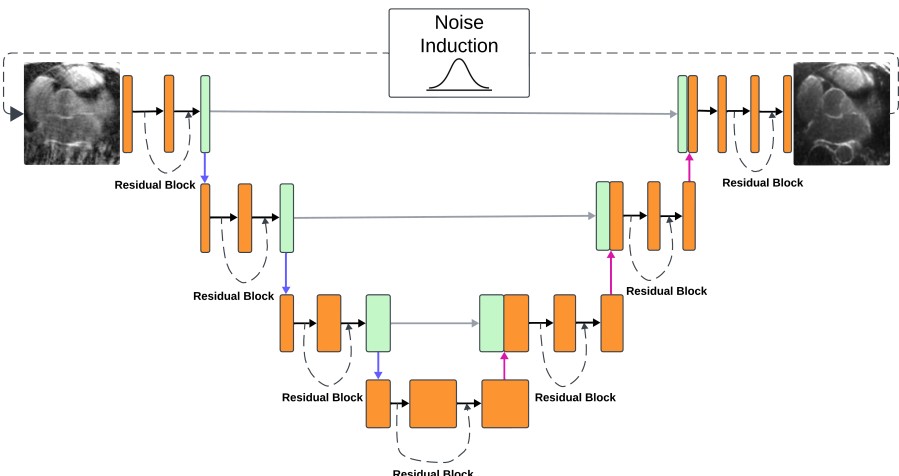

Figure 1: ResUNet architecture with skip connections, residual blocks, and Gaussian noise injection for self-guided DIP denoising.

processing tasks (Kumar et al., 2023). The network processes complex-valued input data by treating real and imaginary components as separate channels. Empirically, we found that Leaky ReLU activation functions demonstrated marginally better performance (PSNR/ SSIM) compared to ReLU or tanh activations. This improvement is likely due to Leaky ReLU's ability to preserve negative values in the complex-valued MRI data, where both real and imaginary components contain important signal information. We used Adam optimizer with learning rate $3 \times 10^{-4}$. For self-guided DIP, we set the denoiser weight $\alpha$ as 4. We found that with $\alpha = 4$, we got the best PSNR. For TV-DIP, we set the regularizarion weight $\lambda_{\text{TV}}$ as 2, as it obtained the best PSNR. For the Gaussian noise $\eta$ in self-guided DIP, we used 0 as mean and $m/2$ as the standard deviation, where $m$ is the maximum value of magnitude of the initial image. We ran all the DIP based methods upto 2500 epochs.

As a baseline comparison, we also implemented a supervised U-Net approach trained on fully-sampled reconstructions (van der Velde et al., 2021). The network was trained using a combination of perceptual loss (based on VGG16 features) and mean absolute error, with the compressed sensing reconstructions from full acquisition time serving as ground truth. We employed a 'leave-one-out' cross-validation strategy, training our model on 9 subjects and testing on the remaining one, iterated across all 10 subjects. Training occured over 1000 epochs using the Adam optimizer with a $3 \times 10^{-4}$ learning rate.

The computational requirements vary significantly across methods. The iterative CS reconstruction required approximately 3.5 hours per volume on a 16-core CPU server with 128GB RAM. The Deep Learning based methods were implemented in PyTorch and ran on an NVIDIA A100 GPU with 80GB memory. Processing times for a full 3D volume (approximately 140 slices) were: Vanilla DIP (28 minutes), Reference-Guided DIP (29 minutes), DIP-TV (37 minutes), and Self-Guided DIP (39 minutes). Memory requirements peaked at 21GB for Vanilla DIP and Reference-Guided DIP, 25GB for DIP-TV, and 35GB for

Self-Guided DIP. For comparison, the supervised U-Net approach required 48GB during training and 7GB for inference.

## 5.2. Experimental Setup

We conducted experiments on cardiac MRI data from 10 subjects, acquired over approximately 12-15 minutes per scan. For each subject, we generated multiple undersampled datasets by extracting k-space data corresponding to different acquisition time fractions (1/6th, 1/4th, 1/2nd, and full time). Motion-consistent data was retained through retrospective navigator gating. Each reconstruction method received identical inputs: the undersampled k-space data, sampling mask, and coil sensitivity maps.

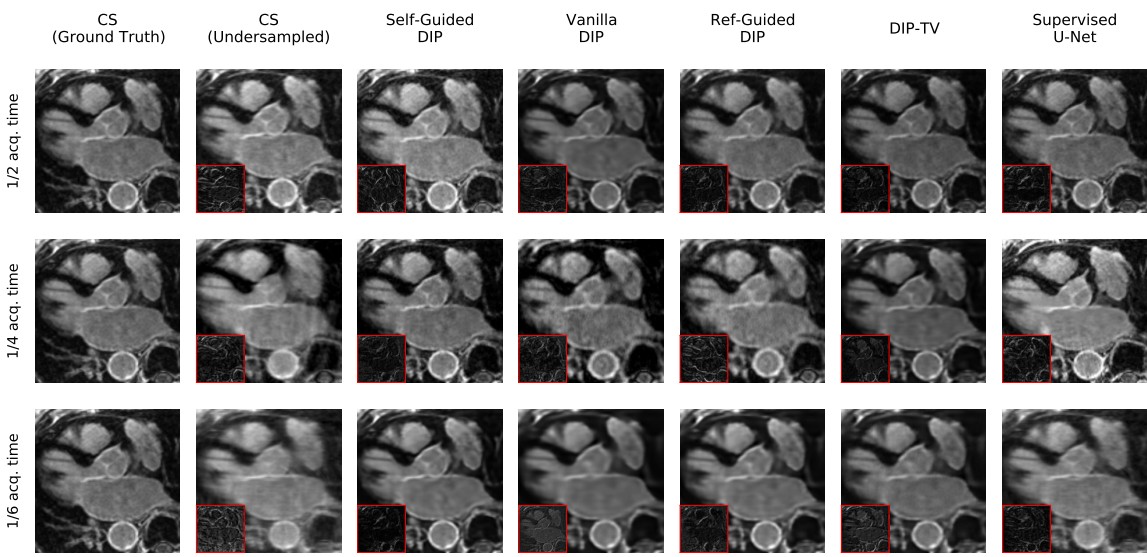

Figure 2: Comparison of LGE-MRI reconstruction methods at different acquisition times (1/2, 1/4, and 1/6 of full scan time), with error maps (red boxes) showing deviations from ground truth.

## 5.3. Analysis

Our evaluation shows Self-Guided DIP's effectiveness for accelerated LGE-MRI reconstruction across different undersampling rates. Figure 2 presents visual comparisons between all methods at varying acquisition times from a randomly selected human subject. We have cropped the images to the heart region for better visualization of the structural details of the region of interest. At 1/4th of the full acquisition time, Self-Guided DIP produces reconstructions visually comparable to fulla acquisition time CS reconstruction, while CS with the 1/4 acquisition time shows significant quality degradation, particularly in the thin left-atrium wall.

For quantitative assessment, we have used two metrics: Peak Signal-to-Noise Ratio (PSNR) and Structural Similarity Index (SSIM). Table 1 presents these metrics averaged

| Method | 1/2 acquisition time | | 1/4 acquisition time | | 1/6 acquisition time | |
|---|---|---|---|---|---|---|
| | **PSNR** | **SSIM** | **PSNR** | **SSIM** | **PSNR** | **SSIM** |
| **Self-Guided DIP** | $34.5 \pm 1.0$ | $0.912 \pm 0.012$ | $32.8 \pm 1.2$ | $0.891 \pm 0.015$ | $29.4 \pm 1.4$ | $0.862 \pm 0.018$ |
| **DIP-TV** | $31.8 \pm 1.3$ | $0.885 \pm 0.015$ | $29.7 \pm 1.5$ | $0.863 \pm 0.018$ | $26.8 \pm 1.8$ | $0.828 \pm 0.023$ |
| **Ref-Guided DIP** | $31.4 \pm 1.4$ | $0.882 \pm 0.016$ | $29.2 \pm 1.6$ | $0.858 \pm 0.020$ | $26.5 \pm 1.9$ | $0.825 \pm 0.024$ |
| **Vanilla DIP** | $29.5 \pm 1.7$ | $0.858 \pm 0.022$ | $27.6 \pm 1.9$ | $0.835 \pm 0.025$ | $24.8 \pm 2.2$ | $0.798 \pm 0.028$ |
| **CS recon** | $30.6 \pm 1.5$ | $0.873 \pm 0.018$ | $28.4 \pm 1.8$ | $0.842 \pm 0.023$ | $25.7 \pm 2.0$ | $0.812 \pm 0.026$ |
| **Supervised U-Net** | $32.3 \pm 1.2$ | $0.889 \pm 0.015$ | $30.1 \pm 1.5$ | $0.867 \pm 0.019$ | $27.2 \pm 1.7$ | $0.835 \pm 0.022$ |

Table 1: Comparison of reconstruction quality metrics (PSNR and SSIM) for different methods at various acquisition time fractions. Values shown are mean ± standard deviation across all subjects.

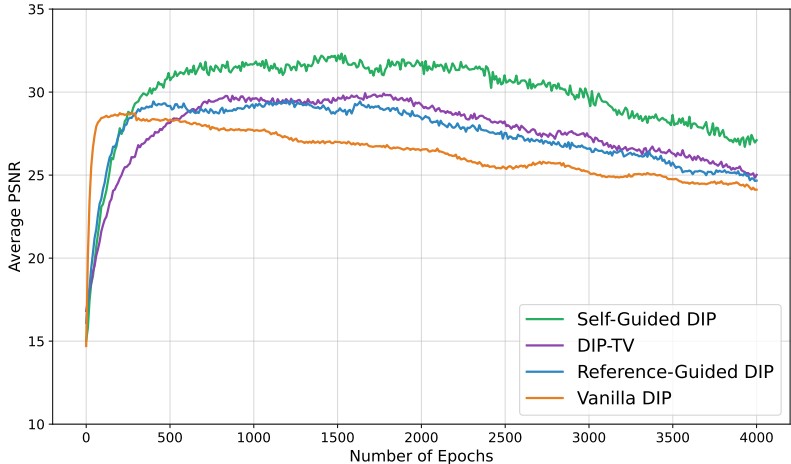

Figure 3: Comparison of average PSNR values across training epochs for different DIP variants. All reconstructions use data from 1/4th of the acquisition time.

across all 10 subjects for each reconstruction method and acquisition time. The values reported correspond to the optimal epoch for each method and subject. These metrics are calculated using the full reconstructed images, not just the cropped heart regions shown in the visual comparisons.

Deep Image Prior models tend to overfit when trained for excessive epochs (Wang et al., 2021). The stability analysis (Figure 3) reveals varying overfitting behaviors across DIP variants. We tested this through an experiment running the DIP models up to 4000 epochs, where PSNR values were calculated and averaged across all subjects at each epoch. This analysis provides insights into the optimal stopping points for different DIP variants. The Self-Guided DIP achieves the highest PSNR and maintains better stability over epochs, but

starts declining around 2300 epochs. While Vanilla DIP shows early convergence but starts declining very soon. Both DIP-TV and Reference-Guided DIP demonstrate intermediate performance with gradual degradation after reaching their peaks around 1700 epochs. The PSNR values were calculated using the full reconstructed images.

An important consideration for working with a prospective subject is determining an appropriate stopping criterion without access to ground truth for PSNR calculation. Based on the experiments above on stability analysis, we find that 2000 epochs generally provide optimal results for self-guided DIP. In future work, we plan to explore metric-based automated stopping criteria, for example, monitoring the data consistency term until it stabilizes or tracking the structural consistency between consecutive reconstructions. Other promising approaches include Stein's Unbiased Risk Estimator (SURE) (Khan et al., 2024) and methods leveraging acquisition noise characteristics, which we aim to explore in our ongoing research efforts.

## 6. Conclusion

This study demonstrates the effectiveness of self-guided Deep Image Prior for accelerated 3D LGE MRI reconstruction, enabling high-quality isotropic imaging $(1.25\text{mm}^3)$ with acquisition times reduced to 2-4 minutes. Our comprehensive evaluation of the DIP variants reveals that self-guided DIP outperforms traditional compressed sensing and other DIP approaches.

This acceleration in acquisition time may address several key clinical challenges by reducing patient discomfort, decreasing motion artifacts, and improving workflow efficiency. This acceleration in acquisition time could potentially facilitate high-resolution 3D LGE imaging, which may improve pre-ablation planning and post-procedure assessment. The method shows particular strength in preserving the details of the thin left atrial wall which is critical to fibrosis assessment.

While our results demonstrate significant promise, several limitations should be acknowledged. First, our evaluation was limited to 10 human subjects, and validation on larger cohorts is needed. Publicly available 3D isotropic LGE-MRI datasets (k-space) for reconstruction tasks are currently unavailable, making broader validation challenging. Second, our study was conducted retrospectively; prospective validation in clinical workflows is necessary to fully assess real-world performance and determine optimal stopping criteria without ground truth references.

Future work should address three key directions. First, integrating explicit motion compensation mechanisms could further improve image quality in regions affected by respiratory motion. Second, comprehensive evaluation across different scanner platforms and sequence parameters would be essential to validate the method's robustness for potential clinical translation. Finally, addressing the fundamental limitations of DIP models - particularly their susceptibility to noise overfitting at high epochs and difficulty in capturing high-frequency features from undersampled data (spectral bias) (Shi et al., 2022) - remains crucial for improving reconstruction quality.

## Acknowledgments

This work was supported by The National Heart, Lung, and blood Institute of the National Institute of Health under award R01HL162353. The content is solely the responsibility of the authors and does not necessarily represent the official views of the National Institute of Health.

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
