# OpenReview forum: "Family of Deep Image Prior Networks for Accelerated 3D LGE-MRI Acquisition with Enhanced Reconstruction"
_MIDL.io/2025/Conference — MIDL 2025 Oral_

### Official Review · Reviewer_bGcY · 2025-02-21

**Confidence:** 3
**Preliminary Rating:** 4
**Recommendation:** Poster
**Final Rating:** 5

**Summary:**

The paper analyzed the effect of Deep Image Prior's implementation for the acceleration of LGE-MRI reconstruction.

**Strengths:**

The paper has a good structure, a clear outcome and is well-written.
The authors provided a comprehensive literature review.
The methods described the proposed solution, results are clear, with visual and quantitive assessment and ablation study.

**Weaknesses:**

No weaknesses are found except for some lack of background and code release.

A limitation section can be added at the end.

Bigger datasets should be used in the future to prove and assess current findings.

**Detailed Comments:**

Figure 3 is Table 3 in reality.

after reaching their peaks around 16700 epochs – don’t match Figure 4

**Justification Of The Final Rating:**

It will be great if authors write the readme for github repo. Overall, the work is brilliant, I sincerely appreciate the replies to my questions which are precise and clear, and the updated paper. Good luck!

**Justification Of The Preliminary Rating:**

The paper is well-written, the motivation is clear and exciting, all methods are described, results are promising, and further directions are drawn. Hopefully, we will see soon the prospective clinical trial.

**Questions To Address In The Rebuttal:**

self-regularization – can authors describe it in more details?

zero-filled reconstruction – what is it, and why can it be counted as a reference?

Can the authors provide a Github link with code?

---

> ### Author Response · Authors · 2025-03-08
>
> We sincerely thank the Reviewer for their positive assessment of our work and for highlighting areas that needed further clarification. We have addressed these important points as follows (added text appears in Cyan in the submitted revision):
>
> 1. **Figure 3 vs Table 3**:
>    We thank the reviewer for noting this discrepancy. We've corrected the reference to identify it as Table 1 throughout the paper properly.
>
> 2. **Epoch value discrepancy**:
>    We appreciate the reviewer for identifying this inconsistency, it was a typing mistake. We've corrected this error to reflect the actual value of "~1700 epochs" where peaks appear in the respective experiments.
>
> 3. **Self-regularization details**:
>    Self-regularization in the self-guided DIP refers to the process where the network output is encouraged to be consistent under small random perturbations of its input. This approach adds stability by acting as an implicit denoising mechanism, effectively preventing overfitting to noise.
>
>    We have added these self-regularization details to the paper.
>
> 4. **Zero-filled reconstruction**:
>    Zero-filled reconstruction is the initial image obtained by directly applying inverse Fourier transform to the undersampled k-space data, filling unsampled locations with zeros. It serves as both a reference point and initialization for our iterative methods. While containing artifacts, it provides structural information that helps guide network optimization.
>
>    We have added these zero-filled reconstruction details to the paper.
>
> 5. **Code availability**:
>    We have made our implementation publicly available on GitHub (https://github.com/inferiorzoned/dips). The repository contains the core reconstruction algorithms and evaluation tools.
>
> 6. **Limitations section**:
>    As suggested, we have added a paragraph in the conclusion section containing the limitations of the current constraints, particularly noting the need for larger datasets and prospective validation.

---

### Official Review · Reviewer_4Gu1 · 2025-02-22

**Confidence:** 3
**Preliminary Rating:** 4
**Recommendation:** Poster
**Final Rating:** 4

**Summary:**

This study examines four different Deep Image Prior (DIP) approaches for accelerating 3D LGE-MRI reconstruction, a technique critical for visualizing and managing left atrial fibrosis. Standard 3D LGE-MRI protocols often require prolonged acquisition times (7–20 minutes) and can yield suboptimal image quality. By contrast, the DIP-based methods evaluated here achieve diagnostic fidelity with scan durations as short as 2–4 minutes, showing that DIP-based methods are suitable for high-resolution 3D LGE studies for atrial fibrillation patients.

**Strengths:**

1. This study introduces a deep learning method capable of super-resolution from ultra-low-field to high-field MRI, reducing scan times substantially (from roughly 7–20 minutes to 2–4 minutes).
2. The authors adapt the DIP framework specifically for MRI reconstruction by integrating MRI acquisition physics, thereby aligning the method with the modality’s inherent constraints.
3. Their use of a 3D deep learning architecture matches the naturally volumetric nature of MRI data, offering better robustness compared to conventional 2D approaches.
4. Experimental protocols systematically vary the acquisition time fractions (1/6th, 1/4th, 1/2nd, and full), demonstrating how the proposed method performs under differing levels of undersampling.

**Weaknesses:**

1. The experiments involve only ten human subjects from a private dataset, which limits the generalizability of the results. Future work would benefit from testing on larger and publicly accessible datasets.
2. While this paper integrates four DIP variants for MRI reconstruction, the most recent among these models was introduced two years prior. From a reviewer’s perspective, it would be valuable to compare against any newer DIP or other state-of-the-art frameworks that have emerged since then.

**Detailed Comments:**

I suggest converting Figure 3 into a properly formatted LaTeX table for clarity and consistency.

**Justification Of The Final Rating:**

This paper addresses an important question regarding the application of several deep image prior networks to Late Gadolinium Enhancement (LGE) MRI. I appreciate the authors’ efforts to address my concerns within the limited time available for rebuttal.

**Justification Of The Preliminary Rating:**

The question the paper wants to solve is important and interesting. Accelerating 3D LGE-MRI reconstruction is essential for improving clinical workflow efficiency and patient comfort. However, the author conducted their experiments on a private 10-subject dataset. Although they have done several experiments with 4 DIP models and two baselines with various acquisition time fractions (1/6th, 1/4th, 1/2nd, and full). From a reviewer's point of view, I wonder whether this amount of experiment is suitable for paper to be accepted in MIDL.

**Questions To Address In The Rebuttal:**

1. I recommend evaluating the proposed method on a larger, publicly available dataset.
2. It would be beneficial to include a comparison with newer DIP-based architectures or additional cutting-edge techniques.

**Special Issue:**

No

---

> ### Author Response · Authors · 2025-03-08
>
> We appreciate the Reviewer's thorough evaluation and highlighting important limitations in our study. We have considered these important points and would like to offer our response as follows (added text appears in Green in the submitted revision):
>
> 1. **Converting Figure 3 into a properly formatted LaTeX table** :
>    We appreciate this suggestion and have implemented it in our revised paper. Figure 3 has been converted to a properly formatted latex table, Table 1.
>
> 2. **Evaluating the proposed method on a larger, publicly available dataset**:
>    We acknowledge the importance of validating our methods on larger, publicly accessible datasets. However, to the best of our knowledge, there are currently no publicly available 3D LGE-MRI datasets designed explicitly for reconstruction tasks. The specialized nature of cardiac LGE-MRI with isotropic resolution, particularly for atrial fibrillation patients, presents unique challenges in data acquisition and sharing.
> While we are actively working with our clinical collaborators to expand our dataset and potentially make a portion available to the research community (subject to institutional approvals and patient privacy considerations), the current 10-subject cohort represents the largest collection of isotropic 3D LGE-MRI data available to our team.
>
>    We have addressed this issue as a limitation in the conclusion section.
>
> 3. **Comparison with newer DIP-based architectures**:
>    We agree that comparing our approach with the latest DIP-based architectures would strengthen the paper. We are currently implementing comparisons with several recent methods, including aSeqDIP [ref: aSeqDIP]. However, these advanced techniques require extensive hyperparameter tuning and optimization to ensure fair comparison, particularly for the challenging 3D cardiac MRI reconstruction task.
> Given the computational intensity of these experiments and our commitment to thorough validation, we propose to include these additional comparisons in our future work.
>
> Ref:
> aSeqDIP: https://proceedings.neurips.cc/paper_files/paper/2024/file/21eba560be81c3a1e1f3404493a92a6a-Paper-Conference.pdf

---

### Official Review · Reviewer_vgWX · 2025-02-24

**Confidence:** 5
**Preliminary Rating:** 4
**Recommendation:** Poster

**Summary:**

This work explores the use of different flavors of Deep Image Prior based models for accelerated late gad enhanced MRI

**Strengths:**

This is a very well written and easy to follow paper that compares a suite of existing ideas in the context of a challenging clinical application. The authors clearly demonstrate the superiority of their proposed method

**Weaknesses:**

************************************I have added a few comments below regarding items that I consider as weakness, please consider addressing them as appropriate.******************************************

**Detailed Comments:**

Define lamba terms in Eqn 1

Please define Nc in Eqn 3

Define the gradient of the CNN in Eqn 4

Define alpha in Eqn 5

The symbol used for norm is incorrect in most of the equations, please fix it

Are the results in Fig 2- 4 corresponding to the best PSNR across the epochs? Please clarify / discuss Fig2-3 in the context of something like Fig-4.

A discussion on stopping criterion is likely needed as prospective scans will not have the benefit of a quantitative metric to guide termination

A brief discussion on the memory and compute needs of the different approaches is needed to allow the reader a better appreciation of the trade-offs involved

**Justification Of The Preliminary Rating:**

Its a well written work with a clear purpose and fairly easy to follow. It also seeks to address the challenges involved in a very important clinical MRI application. I expect the reviewers to be able to address my comments quite readily.

**Questions To Address In The Rebuttal:**

Please consider addressing my comments in your discussion section

---

> ### Author Response · Authors · 2025-03-08
>
> We sincerely thank the Reviewer for the thoughtful feedback and for pointing out important issues that needed clarification. We have addressed all points in our revised paper as follows (added text appears in Red in the submitted revision):
>
> 1. **Lambda terms in Equation 1**: We have clarified the lambda terms (λ₁, λ₂, λ₃) in Equation 1, The revised text now states:
>  "The weights λ₁, λ₂, and λ₃ are carefully tuned using hyperparameter grid search. λ₁ controls the data consistency with acquired k-space measurements, λ₂ controls the spatial smoothness while preserving edges, and λ₃ controls the contribution of non-local self-similarity regularization."
>
> 2. **Nc in Equation 3**: We have defined Nc as:
>    "Nc represents the number of receiver coils used in the MRI acquisition."
>
> 3. **Gradient of CNN in Equation 4**: We have clarified the gradient term with:
>    "where ∇fθ(z) denotes the spatial gradient of the CNN output, which is computed via finite differences in the image domain."
>
> 4. **Alpha in Equation 5**: We have defined alpha as:
>    "where α is the weighting parameter that controls the strength of the self-regularization term, which encourages consistency between the network output and its input under random perturbations."
>
> 5. **Norm notation**: We have corrected the norm notation throughout the equations in the paper, replacing the incorrect |·|₂² with ‖·‖₂² for the L2-norm squared and |·|₁ with ‖·‖₁ for the L1-norm.
>
> 6. **Results in Figures 2-4**:
> Figure 2 represents visual comparisons between all methods from a randomly selected human subject. Table 1 (previously referred to as Figure 3, now correctly identified as a table) presents the quantitative metrics averaged across all 10 human subjects in our experiment. Figure 4 shows the PSNR evolution over epochs, where the PSNR values are averaged across all subjects.
>
>    We have modified the respective paper text accordingly to be more specific and less ambiguous.
>
> 7. **Stopping criterion for prospective scan**: We have added clarification in the Analysis section with the following content:
>
>    "An important consideration for working with a prospective subject is determining an appropriate stopping criterion without access to ground truth for PSNR calculation. Based on the experiments above on stability analysis, we find that 2000 epochs generally provide optimal results for self-guided DIP. In future work, we plan to explore metric-based automated stopping criteria, for example, monitoring the data consistency term until it stabilizes or tracking the structural consistency between consecutive reconstructions. Other promising approaches include Stein's Unbiased Risk Estimator (SURE) and methods leveraging acquisition noise characteristics, which we aim to explore in our ongoing research efforts.”
>
> 8. **Computational requirements**: We have added the following details to the Implementation Details section:
>
>    "The computational requirements vary significantly across methods. The iterative CS reconstruction required approximately 3.5 hours per volume on a 16-core CPU server with 128GB RAM. The Deep Learning based methods were implemented in PyTorch and ran on an NVIDIA A100 GPU with 80GB memory. Processing times for a full 3D volume (approximately 140 slices) were: Vanilla DIP (28 minutes), Reference-Guided DIP (29 minutes), DIP-TV (37 minutes), and Self-Guided DIP (39 minutes). Memory requirements peaked at 21GB for Vanilla DIP and Reference-Guided DIP, 25GB for DIP-TV, and 35GB for Self-Guided DIP. For comparison, the supervised U-Net approach required 48GB during training and 7GB for inference."

---

### Author Rebuttal · Authors · 2025-03-08

**Rebuttal:**

We are grateful for the Reviewer's encouraging comments and constructive feedback. We have addressed the reviewers' concerns in each comment section of respective reviewers, with changes highlighted in colored text in our revised manuscript (Red for Reviewer vgWX, Green for Reviewer 4Gu1, and Cyan for Reviewer bGcY).

We are attaching the revised manuscript as the supporting material here.

**Supporting Material:**

/attachment/10a5e5be327edc0e8b438f6d13ab87d113d439a8.pdf

---

### Author Response · Authors · 2025-03-14

Respected reviewers, as we are approaching the end of the discussion period tomorrow, we wanted to check if you have any remaining questions or if our responses have adequately addressed your concerns.

---

### Meta-Review · Area_Chair_vFti · 2025-03-21

**Recommendation:** Accept (Poster)
**Confidence:** 5

**Metareview:**

All reviewers' ratings are positive following the authors' rebuttal, with one reviewer increasing the score to 'Strong Accept.' The proposed work is important for clinical applications and will add great value to MIDL.